# Synthesis and Physicochemical Characterization of Gelatine-Based Biodegradable Aerogel-like Composites as Possible Scaffolds for Regenerative Medicine

**DOI:** 10.3390/ijms25095009

**Published:** 2024-05-03

**Authors:** Silvana Alfei, Paolo Giordani, Guendalina Zuccari

**Affiliations:** Department of Pharmacy, University of Genoa, Viale Cembrano, 16148 Genoa, Italy; paolo.giordani@unige.it (P.G.); guendalina.zuccari@unige.it (G.Z.)

**Keywords:** tissue engineering (TE), regenerative medicine (RM), copolymer-assisted gelatine crosslinking, lysyl oxidase enzyme (LO), aerogel-like structure, biodegradable scaffolds, high water absorption capacity, high porosity, low density, pseudoplastic behavior

## Abstract

Regenerative medicine is an interdisciplinary field aiming at restoring pathologically damaged tissues and whole organs by cell transplantation in combination with proper supporting scaffolds. Gelatine-based ones are very attractive due to their biocompatibility, rapid biodegradability, and lack of immunogenicity. Gelatine-based composite hydrogels, containing strengthening agents to improve their modest mechanical properties, have been demonstrated to act as extracellular matrices (ECMs), thus playing a critical role in “organ manufacturing”. Inspired by the lysyl oxidase (LO)-mediated process of crosslinking, which occurs in nature to reinforce collagen, we have recently developed a versatile protocol to crosslink gelatine B (Gel B) in the presence or absence of LO, using properly synthesized polystyrene- and polyacrylic-based copolymers containing the amine or aldehyde groups needed for crosslinking reactions. Here, following the developed protocol with slight modifications, we have successfully crosslinked Gel B in different conditions, obtaining eight out of nine compounds in high yield (57–99%). The determined crosslinking degree percentage (CP%) evidenced a high CP% for compounds obtained in presence of LO and using the styrenic amine-containing (CP5/DMAA) and acrylic aldehyde-containing (CPMA/DMAA) copolymers as crosslinking agents. ATR-FTIR analyses confirmed the chemical structure of all compounds, while optical microscopy demonstrated cavernous, crater-like, and labyrinth-like morphologies and cavities with a size in the range 15–261 µm. An apparent density in the range 0.10–0.45 g/cm^3^ confirmed the aerogel-like structure of most samples. Although the best biodegradation profile was observed for the sample obtained using 10% CP5/DMAA (M3), high swelling and absorption properties, high porosity, and good biodegradation profiles were also observed for samples obtained using the 5–10% CP5/DMAA (M4, 5, 6) and 20% CPMA/DMAA (M9) copolymers. Collectively, in this work of synthesis and physicochemical characterization, new aerogel-like composites have been developed and, based on their characteristics, which fit well within the requirements for TE, five candidates (M3, M4, M5, M6, and M9) suitable for future biological experiments on cell adhesion, infiltration and proliferation, to confirm their effective functioning, have been identified.

## 1. Introduction

Tissue engineering (TE) is a fascinating field that aims to create functional tissues or organs for transplantation or regenerative medicine [1], whose name can be traced back to 1988 [2]. However, the foundations of this new field of research, defined as an interdisciplinary field to which the principles of engineering and life sciences are applied, appeared later, in a paper dated to 1993 [3]. TE holds immense promise for treating various conditions, from repairing damaged cartilage to creating artificial organs, and researchers continue to explore innovative techniques and materials to enhance TE outcomes. Essentially, TE involves first the extraction of cells from a tissue in a donor [4]. The types of cell sources used enormously influence TE outcomes. Autologous (the patient’s cells), allogenic (other than the patient’s cells), or xenogeneic (cells derived from animals) cell resources are the three more significant types of cells currently considered in TE and can be derived from various tissues, such as skin, bone, or cartilage [5]. The extracted cells are then cultured in vitro (outside the body) under controlled conditions. This step allows them to multiply and form a cell population suitable for TE [4]. Following this, the cells continue proliferating, dividing, and increasing in number [4]. At this point, to create a new tissue, cells need a proper biocompatible support structure (often called a scaffold), capable of providing a framework onto which cells are seeded, thus allowing them to organize and grow [4,5,6]. Upon the deposition of cells on the scaffold, tissue formation should start and, over time, cells organize themselves, thus differentiating into specific cell types (e.g., muscle cells, bone cells) and forming a functional tissue. Finally, the newly formed tissue is implanted with or without the scaffold into patients requiring intervention through surgery or injection, depending on the tissue type and location. When implanted, the scaffold should ideally resist for the time necessary for the growth and remodulation of the new tissue and then should degrade; it should be resorbed and replaced by the proteins produced by the same cells growing on it [7]. Figure 1, from our recent paper, schematically summarizes the main steps of TE reported here [8]. Using TE, it is possible to have available effective substitutes for the liver, kidneys, and pancreas, as well as for tissues such as blood vessels, skin, cartilage, bones, ligaments, and tendons. Using TE, long-standing commercial skin substitutes have been developed, and bladders have been reconstructed in the past [9,10]. More recent achievements include the replacement and repair of cardiac or nervous tissue [11,12,13], the repair of bone defects [13,14], and bone and dental bone regeneration [15]. Due to the great potential of TE, incessant research is needed in this field to further improve the already considerable advances made in regenerative medicine, mainly in terms of the scaffolds’ properties and resources as well as of cell sources, which now also include stem cells [4,16]. Support scaffolds are crucial for a successful regenerative therapy, with both the function of providing mechanical support to the cells that are forming the new tissue and determining the dimensions and shape of the tissue itself [7]. An ideal support for TE should be primarily biocompatible and should not induce immune responses, thus avoiding undesirable side effects such as inflammation. Otherwise, it should also possess the proper mechanical properties [7], mouldability, and biodegradability after its implantation without producing toxic metabolites [7]. Scaffolds for TE should be capable of performing the typical activities of the extracellular matrix (ECM), such as supporting cell adhesion and allowing the diffusion of nutrients, metabolites, and growth factors [7]. Up until today, different types of materials, including inorganic materials such as metals [17] and ceramics [18,19], natural polymers, synthetic non-biodegradable polymers, biodegradable synthetic polymers, polyphosphazenes (PZs), bio-resorbable polymers (poly-D, L-lactide), bio-erodible polymers, and bio-absorbable polymers, have been applied to prepare promising scaffolds for TE [20,21,22,23,24,25]. However, the protein materials composing the ECM, including collagen and elastin, are the most appealing materials, due mainly to their capability to generate 3D hydrogel structures [26]. Recently, materials derived from decellularized ECM (dECM) have been extensively investigated in TE and regenerative medicine [7]. dECM preserves the native tissue’s composition, both in terms of its structural proteins, such as collagen, and of its growth factors and cytokines, which can improve cell growth and viability and tissue repair and remodeling [27]. Additionally, using different processing methodologies, dECMs suitable for TE have been obtained from a variety of tissues, such as bone, cartilage, meniscus, tendons, skin, adipose tissue, urinary bladder, small intestinal submucosa, liver, and brain [28,29,30,31,32,33]. Collagen and gelatine are extensively applied in several sectors, including biomedicine [10,34,35,36,37]. However, while collagen is an expensive material that can also induce immunogenic responses, gelatine is a low-cost collagen derivative [38] which can provide hydrogels that have been used in the vascularization of engineered tissues without triggering undesired immunogenic responses. Unfortunately, both collagen and gelatine have demonstrated modest mechanical properties, such as low stiffness and a high biodegradability rate, which limit their use in the preparation of efficient scaffolds for TE [39,40,41,42,43,44]. Chemical reticulations, promoted by different crosslinking agents, represent a successful approach to addressing the mechanical issues of collagen and gelatine. Several crosslinking agents and procedures have been used to crosslink both collagen and gelatine, each one endowed with advantages and drawbacks [45,46,47,48,49,50,51,52,53,54,55,56,57,58,59,60,61]. Our first contribution to the research on TE consisted of developing a novel protocol to crosslink gelatine without resorting to the commonly used crosslinking agents, but mimicking natural enzymatic processes. In this context, collagen is naturally stabilized and strengthened by crosslinking reactions through the lysyl oxidase (LO)-assisted oxidation of the amino groups of its lysine and hydroxylysine residues to aldehyde groups. Subsequent aldolic condensation reactions or imine bond formations between the oxidized residues or between the oxidized groups and amine residues, respectively (Scheme 1, in Alfei et al., 2024), lead to a mechanically improved reticulated framework [8].

In this example and based on our previous studies on LO [62], we have recently prepared three amine-containing hydrophilic copolymers, namely CP5/DMAA, CP11b/DMAA, and CP11c/DMAA (Figure 1), whose primary amine groups mimic those of the lysine residues of collagen (Scheme 1, Alfei et al., 2024), thus resulting in substrates for LO [8]. The enzymatic oxidation of their amine groups afforded the desired aldehyde functions, which promoted the reticulation of Gel B by the processes illustrated in Schemes 2 and 3 in Alfei et al., 2024 [8]. In addition to these amine-containing copolymers, a copolymer already containing this aldehyde functionality, synthesized by commercially available methacrolein (MA) (CPMA/DMAA, Figure 1), was also used in experiments of the crosslinking of Gel B, and also in the absence and presence of LO [8].

**Figure 1 ijms-25-05009-f001:**
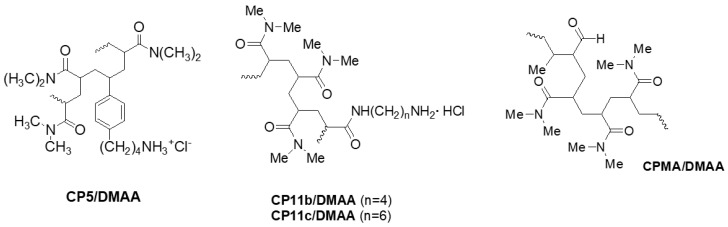
Copolymers prepared according to Alfei et al. [8,63], which were successful in crosslinking Gel B. CP = copolymer; **5** = aminobutyl styrene hydrochloride; **11b** = *N*-acryloyl-1,4-diaminobutane hydrochloride; **11c** = *N*-acryloyl-1,6-diaminohexane hydrochloride.

Here, following a slightly modified protocol, we have expanded these crosslinking experiments by reticulating Gel B in further different conditions, using the copolymers shown in Figure 1. Nine compounds (M1–M9), in the form of aerogel-like composites, were achieved after lyophilization. The eight gained in a satisfying yield were titrated to assess their crosslinking degree percentage (CP%), and the seven with an acceptable CP% (M1–M6 and M9) were subjected to attenuated total reflection Fourier transform infrared (ATR-FTIR) spectroscopy analyses to qualitatively confirm their structure, as extensively suggested in the literature [64,65,66,67,68,69,70,71]. Although we are aware of the higher precision of SEM and AFM techniques in characterizing the morphology and topography of micro- and nanomaterials, to reduce the study costs, before resorting to these techniques which are not available in our laboratory, optical microscopy was used to investigate the morphology and pore sizes of the prepared crosslinked compounds, as also reported in the literature [72,73]. Further experiments were then carried out to verify their possible suitability for future experiments on cells’ adhesion and proliferation. To this end, we determined their equilibrium swelling rate (%), water absorption capacity (%), apparent density and porosity (%), equilibrium water content (EWC%), biodegradability, water loss rate (%), and rheological properties, thus identifying M4, M5, M6, and M9 as the most promising compounds for future biological experiments. Among them, M4, M5, and M6, produced using CP5/DMMA copolymers, previously reported to be non-cytotoxic [8,74], could be the most biocompatible choices.

## 2. Results and Discussion 

### 2.1. Gelatine Crosslinking

#### 2.1.1. Commercially Available Gelatine

The gelatine used in this study was type B gelatine (Sigma-Aldrich, Darmstadt, Germany), hereinafter referred to as Gel B, whose main characteristics have been previously reported [8]. The determination of the amino groups in Gel B was performed both via acid–base titration, following an experimental procedure applied to Gel B-type gelatines [75], and spectrophotometrically [8]. It was found that the NH_2_ equivalents, by acid–base titration, were 0.189 mmol/g (pH = 8.5) and 0.593 mmol/g (pH = 11.5), thus creating the average value of 0.391 mmol/g. Contrastingly, the NH_2_ equivalents by spectrophotometric determination were 0.219 mmol/g. The lower value obtained in this case, with respect to that measured by titration with NaOH, may be due to an incomplete functionalization of the lysine residues of Gel B necessary for this determination, thus providing an amount of chromophore molecules that is inferior to the actual number of amine groups [8]. 

#### 2.1.2. Gelatine Crosslinking 

Using the NH_2_-containing copolymers, CP5/DMAA, CP11b/DMAA, and CP11c/DMAA, that generated a positive Schiff test in the preliminary experiments on enzymatic oxidation [8] and the aldehyde-containing copolymer CPMA/DMAA shown in Figure 1, Gel B was reticulated in presence or absence of LO, previously extracted by us from bovine aorta or PAO (Worthington, Biochemical Corporation, Lakewood, NJ, USA), according to Figure 1 [8].

**Scheme 1 ijms-25-05009-sch001:**
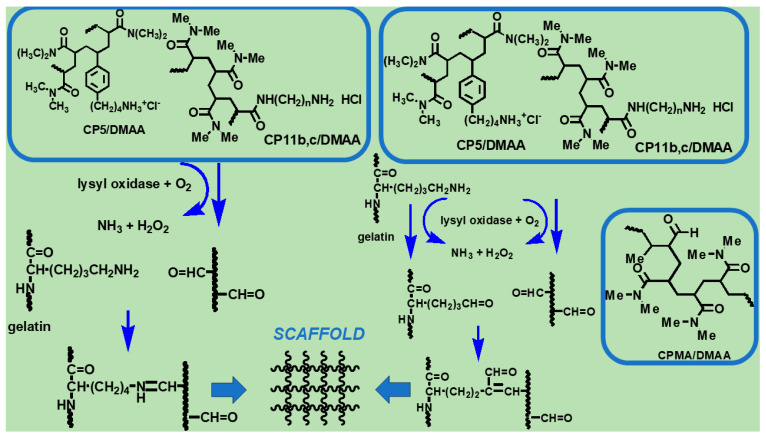
Reactions that can occur during the LO-assisted crosslinking of gelatine [8].

Unlike what has been previously reported [8], the samples were isolated from the aqueous medium upon congelation (−70 °C) and lyophilization, instead of simple desiccation. Appendix A shows the appearance of the most representative samples of crosslinked Gel B. The experimental data and results concerning the performed reticulation reactions are included in Table 1.

#### 2.1.3. Titration of Crosslinked Gelatines

The equivalents of the residual NH_2_ not reacted during the reticulation reactions were determined by a spectroscopic method and/or by acid–base titration, as previously described [8]. All experiments were performed in triplicate, and the results were expressed as mean ± standard deviation (SD). M8 was not titrated due to its extremely low yield. The percentage of crosslinking (CL%) of the considered samples (M1–M7 and M9) was calculated according to Equation (1).
CL% = {[(mmol NH_2_) _Gel B_ − (mmol NH_2_) _Gel B CL_]/(mmol NH_2_) _Gel B_} × 100(1)

In (1) (mmol NH_2_) _Gel B_ is the NH_2_ (mmol/gram) contained in pristine Gel B and (mmol NH_2_) _Gel B CL_ is the NH_2_ (mmol/gram) contained in crosslinked Gel B. The results are included in Table 2.

For sample M7, a very low crosslinking value was recorded both by UV and NaOH titration, and therefore the sample was no longer considered in the following analyses. Contrastingly, the other samples demonstrated crosslinking percentages, by acid–base titration, in the range 29–60%. Except for sample M3, whose UV titrations provided a crosslinking percentage higher than that obtained by NaOH titrations, all other samples provided UV titration results significantly lower than those obtained by the acid–base method, as observed also for pristine Gel B. The UV determinations of sample M2, prepared without the enzyme and with the methacrolein/DMAA copolymer (CPMA/DMAAA-20), provided a crosslinking percentage close to zero. However, it should be considered that, in this sample, the most probable crosslinking mechanism was one via imine-type bond formations, which could be unstable in the hot HCl treatment necessary for UV titration. Consequently, it could be assumed that the crosslinking bonds could have been destroyed during the UV analysis, thus causing the decrease in absorbance values close to those of non-crosslinked Gel B. With these considerations, titration with NaOH has to be considered the more suitable method to evaluate the crosslinking degree in such samples (59.8%). Interestingly, sample M9, obtained by crosslinking Gel B with the CPMA/DMAAA-20 copolymer, but in the presence of LO, showed a decidedly higher crosslinking percentage (46.1%) than sample M2 (5.5%). These findings established the important role of the enzyme in the formation of crosslinked structures of gelatin via non-hydrolysable bonds.

#### 2.1.4. ATR-FTIR 

To detect the functional groups in the crosslinked gelatines and qualitatively assess their chemical structure and composition, ATR-FTIR spectra were acquired from samples which demonstrated a proper crosslinking degree (M1–M6 and M9), as suggested in the literature [64,65,66,67]. The ATR-FTIR spectra of pristine Gel B and of the copolymers CPMA/DMAA and CP5/DMAA were also acquired for comparison and have been shown in Appendix A. In particular, the FTIR spectrum of Gel B showed a C-N stretching band at 1237 cm^−1^, the C=O stretching band of the amide II and of the amide I at 1531 cm^−1^ and 1632 cm^−1^, respectively, -CH_2_ bending and -CH_2_ stretching at 1446 cm^−1^ and 2938 cm^−1^, respectively, and, finally, the N-H stretching of amide I and II was detected at 3280 cm^−1^ in the form of a broad band [76]. In the spectrum of copolymer CPMA/DMAA (abbreviated to CPMA in Appendix A), a band of C-H stretching was observable at 2930 cm^−1^, while bands of aldehydic and of amide C=O stretching were observable at 1720 cm^−1^ and 1611 cm^−1^, respectively [8]. Otherwise, in the spectrum of copolymer CP5/DMAA (abbreviated CP5 in Appendix A), we observed significant bands at 3470 cm^−1^ (NH_3_^+^) and at 1620 cm^−1^ (C=O) [63]. Except for the band at 1720 cm^−1^, due to the C=O stretching of the aldehydic group present in the structure of copolymer CPMA/DMAA, the spectra of the two copolymers were very similar due to the high concentration of DMAA in their composition (80% in CPMA/DMAA and 50–95% in CP5/DMAA). Figure 2 shows the spectra of all crosslinked samples and that of Gel B.

From an early observation, the spectra of all crosslinked samples (colored plots) in which all Gel B bands were well detected were very similar to that of Gel B (Figure 2). Moreover, due to the low concentration of MA monomer in the CPMA/DMAA copolymers (20%) and of copolymer in the crosslinking mixture (10%), the band of the aldehydic C=O stretching in samples M2 and M9 was no longer visible. The spectral data of all analyzed samples were then arranged in a 3401 × 10 matrix of 34,010 measurable variables and were processed using a principal components analysis (PCA) by means of CAT statistical software (16 April 2024) (Chemometric Agile Tool, free down-loadable online, at: http://www.gruppochemiometria.it/index.php/software/19-download-the-r-based-chemometric-software; accessed on 29 April 2024). For each sample, the variables consisted of the values of transmittance (%) associated with the wavenumbers (3401) in the range 4000–600 cm^−1^. The spectral data in the matrix were pretreated with autoscaling before PCA. The PCA allowed us to reduce the high number of correlated variables (34,010), also including those variables providing non-significant and redundant information, to a limited number of uncorrelated variables, namely principal components (PCs). The results from the PCA have been reported as a score plot of PC1 (explaining 68.7% of the variance) vs. PC2 (explaining 22.4% of the variance), where the scores are the new coordinates of the processed samples in the new space of the PCs Appendix A. Within the score plot, each sample assumed a position depending on its chemical composition and structure. Samples located close to each other shared similar characteristics, while those placed far apart differed. As observable in Appendix A, except for M9, the crosslinked samples were located all in the upper half of the plot at scores positive or close to zero (red circle), while non-crosslinked Gel B and the copolymers were found to have negative scores (green circle). The unexpected position of M9 in a negative area could indicate the presence of residual unreticulated material probably due to defective dialysis. Despite this, all gelatine-containing samples were grouped close to Gel B (light blue square) thus indicating their structural similarities, as previously demonstrated by the bands in the spectra in Figure 2. The copolymers were located very far from each other and at the opposite sides on PC1 (CP5/DMAA had a positive score, and CPMA/DMAA a negative score), confirming the presence of different functional groups in their structure. In PC1, samples M1, M4, and M5 were positioned as having positive scores, like that of CP5/DMAA, probably due either to the high concentration of polystyrene monomer in their copolymerization reaction (M1) or to the higher concentration of styrene copolymer in their crosslinking reactions (M4 and M5). Otherwise, at negative scores in PC1, closer to CPMA/DMAA, we found the samples derived from reactions carried out with lower concentrations of styrene-based compounds and from those carried out with CPMA/DMAA in place of CP5/DMAA. 

#### 2.1.5. Morphology of Crosslinked Gelatines

The morphology of the prepared crosslinked gelatines was investigated by optical microscopy (OM) analysis [72,73], which allowed us also to determine their average pore size (Figure 3, Figure 4 and Appendix A). To this end, three to nine cavities were measured by the instrument for each sample, and their size was expressed as mean ± SD.

Samples M1 (Figure 3a,b) and M2 (Figure 4a,b), although obtained using different crosslinker agents, demonstrated a similar cavernous morphology, which displayed non-uniform large cavities compared to the other samples. An average size of 261.5 ± 157.8 µm was determined for M1 and 225.1 ± 70.2 µm for M2. These findings established that M1 and M2 could be promising scaffolds for mesenchymal stem cells’ proliferation, since it has been reported that they better multiply on supports with pores of a size > 94 and 130 µm, and close to 300 µm, because of the better nutrient accessibility [77]. Interesting, since, as discussed in the following Section 2.1.6, these samples also demonstrated a similar cumulative swelling rate (%), a correlation between their morphology and swelling capacity could be supposed. Samples M4 (Figure 4b) and M6 (Figure 4c) showed a similar crater-like morphology, displaying, in this case, a high number of very small cavities with an average size of 18.2 ± 6.2 µm (M4) and 17.8 ± 10.3 µm (M6). Also, in this case, in the equilibrium swelling rate experiments, M4 and M6 were the samples that had the highest water absorption capacity (WAC%) values, thus confirming that morphology is correlated to the swelling behavior of the samples. Particularly, a lower number of large cavities corresponded to low WAC% values, while a high number of small cavities corresponded to high WAC% values. A marginally different morphology with respect to that of M4 and M6, displaying slightly larger cavities, was demonstrated by sample M9 (Figure 4d), whose average pore size was 93.1 ± 38.6 µm. A singular morphology was instead demonstrated by M3 (Figure 4a) and M5 (Appendix A). In this case, the optical images showed labyrinth-like microstructures with some small cavities, whose average size was assessed as 15.2 ± 5.4 µm (M3) and 15.5 ± 4.9 µm (M5). From these results, sample M9 could be suitable for biological experiments with human skin fibroblasts, since it has been reported that their viability is improved on materials with a pore size around 70–160 µm [78]. As reported, the proliferation speed of cells in small pores is three times higher than that of cells in large pores, thus establishing that samples M3, M5, M4, and M6 could be very promising for supporting rapid cell proliferation [79]. Collectively, due to the dimensions of their cavities, M3, M4, M5, M6, and M9 could support several cellular activities such as cell proliferation, osteogenesis, adipogenesis, cell infiltration, and skin regeneration [80]. Concerning M9, in addition to the dimension of its cavities, its equilibrium swelling rate value was higher than those of M1 and M2 and lower than those of M4 and M6, thus further confirming the existence of a correlation between the morphology, pore size, and swelling behavior of these samples. Curiously, concerning M3 and M5, despite their similar morphology and pore sizes, in swelling experiments they demonstrated very different behaviors.

#### 2.1.6. Equilibrium Swelling Rate

Swelling determinations were made at fixed time points, as previously described [81,82]. All experiments were performed in triplicate, and the results were expressed as the mean ± SD. The swelling experiments provided samples in the form of hydrogels. The cumulative swelling rate (%) over time was calculated and the obtained data were plotted vs. time, obtaining the curves in Figure 5. 

The equilibrium swelling rate (*Q e*) was determined at that time point after which the swelling rate (%) did not further improve. Graphically, *Q e* values corresponded to the intercepts on the y axis of the tangents to the plateau of the curves in Figure 5. The experimental *Q e* values have been indicated in Figure 5 for each sample. Sample M4 reached its *Q e* (1250%) more rapidly than all other samples, after 90 min, followed by M1 and M3, which reached their *Q e* after 120 min. Samples M2, M5, M6, and M9 reached their *Q e* after 150 min, and very similar values (242%, 2710%, and 751%) were reached by M2, M6, and M9 after smaller times of 90, 120, and 120 min, respectively. Compared to other hydrogels reported to be promising for applications in TE, our samples demonstrated similar or higher *Q e* values [83]. Particularly, samples M4, M5, and M6 showed a maximum swelling 2.9–6.3 times higher than that showed by the best sample reported in other studies, with M6 being the sample that demonstrated the highest capacity to absorb water and swell over time. 

##### Kinetic Studies

The kinetics governing the water adsorption by the prepared aerogel-like composites were assessed by carrying out a kinetic study, as previously reported [84,85,86]. To this end, the data on the cumulative swelling ratio plotted in Figure 5 were fit with pseudo-first order (*PFO*) [Equation (2)], pseudo-second order (*PSO*) [Equation (3)], and Peleg’s kinetic models [Equation (4)].
(2)$$\ln\left(Qe-Qt\right)=lnQe-K(PFO)\times t$$
(3)$$\frac{t}{Qt}=\frac{1}{K(PSO)\times Qe^{2}}+\frac{1}{Qe}t\;$$
(4)$$\frac{t}{Mt-Mo}=K1+K2\times t$$
where *Q e* (%) and *Q t* (%) were the swelling (%) at equilibrium and at time *t*, respectively, *Mo* (mg) and *Mt* (mg) were the mass of samples at time *t*_0_ and time *t*, respectively, *K* (*PFO*) was the adsorption constant of the PFO kinetic model, and *K* (*PSO*) was the equilibrium constant velocity of the PSO kinetic model, while *K*1 and *K*2 were the Peleg’s rate and capacity constants. *K*1 is correlated to the adsorption rate (*R*_0_) at the beginning (*t = t*_0_) according to the relation *R*_0_ = −1/*K*1. The values of ln *(Q e-Qt)*, *t/Qt* and t/(Mt−Mo) were plotted vs. time. The obtained dispersion graphs Appendix A were processed by Microsoft Excel software using the Ordinary Least Squares (OLS) method, and their linear regression lines were obtained. As reported, the coefficients of determination (R^2^) of all the equations (Appendix A reported in Appendix A were the parameters for determining the kinetic model that best fit the experimental data of the water absorption processes [87]. All samples best fit the Peleg’s kinetic model. Peleg’s rate and capacity constants were obtained from the intercepts and slopes, respectively, of the equations reported in Appendix A and are included in Table 3.

In particular, *K*1 represents the rate at which water is absorbed by the aerogel-like composites during soaking, and its value depends on temperature and can vary significantly. On the other hand, *K*2 is not temperature-dependent. Essentially, *K*2 indicates the gelatines’ capacity to hold water. According to their *K*1 values, M5 was the sample with the lowest absorption rate, followed, in order, by M1, M2, M3, M4, and M6. M9 was the sample with the highest absorption rate. Concerning *K*2, the lowest values indicated the samples with the highest *Q e*. Using Equation (4), the mass of all samples at their equilibrium was computed, which allowed us to calculate the values of *Q e*
_MODEL_ reported in Table 3. According to the last column of Table 3, the error (%) between the experimental and computed *Q e* values was, in all cases except for M6, <1%, thus confirming that Peleg’s equation was suitable to model the obtained absorption experimental data. The higher error obtained for M6 is consistent with its lower value of R^2^, with respect to other samples. 

#### 2.1.7. Water Absorption Capacity (WAC%), Apparent Density, and Porosity (%)

The WAC% of the crosslinked gelatines, understood as the maximum swelling, and porosity (%), understood as the equilibrium water content (EWC%), were determined as described in Section 3, and the results are included in Table 4. All experiments were performed in triplicate, and the results were expressed as the mean ± SD.

Additionally, porosity (%), defined as the percentage of void space in a 3D sponge, was determined using a gravimetric method according to the formula reported in experimental Section 3 [88]. The apparent density (ρ*) of all samples was also determined to calculate their porosity (%) [89] and to confirm their aerogel-like structure (Table 4). The WAC results agreed with the results of maximum swelling, obtained as the equilibrium swelling rate (*Q e*) in previous experiments. Higher discrepancies were detected for samples M1, M5 (errors of 11% and 10%, respectively), and M9 (7%), while M2, M3, M4, and M6 gave an error of <4%. The plot of the values of WAC% vs. those of porosity (EWC%) evidenced their logarithmic correlation (Appendix A. Concerning WAC, M6 demonstrated a capacity to absorb water higher than that of two out of four nanocellulose fibers (NCFs)/collagen aerogels which have a proved capacity to support cell growth and proliferation [90]. According to Table 4, in the third column, the apparent density values calculated for all samples were in the range 0.1–0.45 g/cm^3^. According to a study by Greco et al., aerogels are materials known for their extremely low density, which, by definition, should be lower than 0.5 g/cm^3^ [91]. In this regard, the results of apparent density reported in Table 4 confirmed the aerogel-like structure of all the prepared samples, with M4, M6, and M9 being the aerogels with the lowest apparent densities. However, according to another more restrictive study by Noman et al., the density and porosity of aerogels should range between 0.040 and 0.350 g/cm^3^ and 85.0 and 99.9%, respectively [92]. With these different data, only M4, M6, and M9 should be considered aerogels. The porosity (%) results obtained by two different methods were sufficiently in agreement. Particularly, the EWC values of M4, M5, M6, and M9 were like those reported for FS hydrogels described to have potential as scaffolds for TE [83]. Additionally, except for M1, all other samples showed EWC% values higher than those of FRH-PG and FRH-PGS scaffolds, which, in in vivo implantation into full-thickness rat patellar tendon defects, showed good collagenous tissue ingrowth [93]. Collectively, it has been reported that scaffolds with 60–90% porosity are suitable for wound healing applications, as they are capable of providing sufficient space for cell activity, oxygen and nutrient exchange, and the production of a new extracellular matrix (ECM) [94]. On these considerations, all samples developed here could be suitable for promoting cells’ attachment and proliferation.

#### 2.1.8. In Vitro Evaluation of the Biodegradability of Crosslinked Gelatines over Time by Mass Loss Experiments

Biodegradability is an essential property that materials finalized for biomedical applications should possess [95], especially those ones designed as scaffolds for TE, that, when used in vivo, should be reabsorbed by the same tissue they generated. However, a too rapid degradation is also not desirable, because it would not ensure the time necessary for cell proliferation and the remodeling of the tissue [96]. To evaluate the biodegradability of the prepared samples over time in solution, we monitored their mass loss by recording their weight change after a specified time of incubation in phosphate-buffered saline (PBS), at 37 °C. To this end, we followed a slightly modified version of a procedure previously reported, as described in Section 3 [97]. Gel B was analyzed in parallel in the same conditions, for comparison purposes. The cumulative mass loss percentages (ML%) over time were plotted vs. time, expressed in days, generating the lines in Figure 6. All experiments were performed in triplicate, and the results were expressed as means ± standard deviation (SD).

In these experiments, we investigated the degradation of Gel B due to its progressive absorption of water and the consequent hydrolysis of its linear protein chains. Concerning crosslinked samples, we monitored their degradation over time, due to their progressive absorption of water and consequent hydrolysis of the imine bonds formed during crosslinking, thus leaving free the linear protein chains of Gel B to be hydrolyzed as well. As observable in Figure 6, pristine Gel B was demonstrated to be the most degradable material, with only 4% of polymer remaining and 96% of its weight lost in two days. Its crosslinking with CP5/DMAA (10%) (sample M5) reduced its max degree of degradation from 96 to 88% and increased the time necessary to reach this max degradation percentage (8 days). Samples M1, M4, and M2, obtained by crosslinking Gel B with either P5/DMAA in different concentrations (M1, M4) or CPMA/DMAA (M2), all reached a similar max percentage of degradation, with 14–19% of polymer remaining; 84, 81, and 86% of their weight lost; and different mass loss profiles, within 3 days. A lower degree of degradation was observed for samples M6 and M9, obtained by crosslinking Gel B with CP5/DMAA (5%) and CPMA/DMAA, respectively. Both reached similar max degradation percentages of 56 and 62% after 3 and 2 days, respectively, and did not lose further weight over subsequent days. This behavior established that a higher percentage of reticulation (based on acid–base titrations) could confer to materials a higher resistance to hydrolytic degradation. The most promising mass loss profile appeared to be that of M3, which, among the crosslinked materials, reached the highest percentage of degradation (90%) with only 10% residual polymer remaining, but after 6 days. This behavior, in a possible future application of M3 for regenerative medicine, could give cells more time to proliferate, while its almost full degradation could allow for its easier reabsorption by tissues. The biodegradability of our samples was compared with that of the multifunctional carboxymethyl chitosan and oxidized cellulose biodegradable (COB) hydrogel scaffolds developed by Shengyu Li et al. [98], who monitored the weight loss of their samples for four days. Concerning M3, its degradation percentage (4 days) was lower than that of all samples reported in their study, but it was almost twice as high after eight days. Furthermore, the maximum biodegradation of M6 and M9 was just like that of all COB hydrogels, while that of M1, M2, M4, and M5 was higher [98]. Since the COB hydrogels of Shengyu Li et al. were demonstrated to be suitable for TE applications [98], we can conclude that our samples M6 and M9, with similar biodegradation profiles, could be suitable for further investigations in cell adhesion and proliferation experiments. As reported above, the degradations of crosslinked gelatines studied here were mainly due to hydrolysis. In this regard, a higher adsorption rate and/or capacity should be correlated to a higher degradation rate and mass loss. On the contrary, M6, which showed the highest water absorption capacity, and M9, which demonstrated the highest adsorption rate, had the lowest degradation and mass loss. This fact can be explained, assuming that, during the crosslinking process, aldolic condensation reactions leading to non-hydrolysable bonds were preferred to the formation of hydrolysable imine bonds, thus providing materials that were less degradable, although capable of rapidly absorbing water. The mass loss profiles of the crosslinked gelatines were further studied by fitting the experimental data with the most common mathematical kinetic models to investigate the main mechanism which regulated their water absorption, hydrolysis, and consequent degradation. The adopted kinetic models included the first-order, the pseudo-second-order (PSO), the Hixson–Crowell, the Higuchi, and the Korsmeyer–peppas models [87,99]. The dispersion graphs, their associated linear regression lines, and related equations are shown in Figure 7 and Figure 8 and in Appendix A. The linear regressions and related equations were provided by Microsoft Excel software 365 using the OLS method. According to the literature, the coefficients of determination (R^2^) associated with all the equations (Appendix A) were considered the parameters used to determine which model best fit the mass loss experimental data [87]. While M3 (fuchsia lines, Figure 7 and Figure 8 and Appendix A best fit the Korsmeyer–peppas model (R^2^ 0.9260, Figure 6), all other crosslinked gelatines best fit the PSO kinetic model (Figure 7).

These results confirmed that M3’s mass loss and degradation behavior were strongly different from those of all other samples. In particular, Korsmeyer–peppas kinetics are described by Equation (5) [95].
(5)$$\ln\left(ML\%\right)\;t=n\times \ln\left(t\right)+\ln\left(K\right)$$

In (5), (*ML*%) *t* is the cumulative mass loss (%) at time *t*, with *ln* (*ML%*) *t* and *Ln* (*t*) being the variables *y* and *x* in the equation of M3 in Figure 7, respectively. *K* is the transport constant, which incorporates structural/geometrical information about the polymer system, while *n* (also called the diffusional or transport exponent) provides information on the possible mechanism(s) of mass loss and degradation [100]. Particularly, *n* corresponded to the slope of the equation of M3 shown in Figure 7 (fuchsia), while *Ln* (*K*) was its intercept. From these calculations, *K* was positive and equal to 6.19 (Table 5), while the value of *n* was 1.26 (Table 5). 

Since *n* was >0.89, it was established that the mass loss from M3 followed a super case II transport mechanism [101].

As for all the other samples, the PSO kinetic model is described by Equation (6).
(6)$$\frac{t}{Wt}=\frac{1}{K\times We^{2}}+\frac{1}{We}t\;$$

In (6), *W e* and *W t* are the weight of the samples at their maximum degradation and at time *t*, respectively, while *K* is the equilibrium constant velocity of the PSO kinetic model. Particularly, PSO dispersion graphs were obtained by plotting the values of *t*/*W t* vs. time. 

In processes ruled by PSO kinetics, chemical reactions involving the sharing or exchange of electrons between the degrading materials and water, or electrostatic interactions, are the main mechanisms by which water interacts with the sample during degradation [85,86].

The *K* and *W e* values provided by the mathematical model (*We*
_PSO_) were computed using the values of the slopes and intercepts of the equations in Figure 8 for all samples (except for M3) and are included in Table 5.

For all samples following the same mathematical model, a higher value of *K* corresponded to a higher rate of degradation. The results in Table 5 confirmed the assumptions previously reported during the simple observation of mass loss profiles reported in Figure 6. Gel B was the sample with the highest rate of degradation, followed by M5 and M9, and in turn followed by M6, M2, and M1, which demonstrated similar rates of degradation. Among the samples degrading with PSO kinetics, M4 showed the lowest rate of degradation. Additionally, the calculated value of *We* for all samples following PSO kinetics perfectly agreed with the experimental ones, thus establishing that the diffusion stage within the materials was not involved in the mechanism of the degradation of these samples [102]. The reticulation and degradation results were analyzed together by arranging the data concerning the maximum weight loss (%), the time to reach it, the degree of crosslinking by acid–base and UV titration, and the values of constants *K* in Table 5 in a 5 × 8 matrix of 40 correlated variables. Then, as performed for the ATR-FTIR analyses, to obtain the most significant information from these data and eliminate those which were not substantial or redundant, these 40 variables were reduced to three uncorrelated PCs, the constructed matrix was processed by PCA using CAT (16 April 2024), (Chemometric Agile Tool, free down-loadable online, at: http://www.gruppochemiometria.it/index.php/software/19-download-the-r-based-chemometric-software; accessed on 29 April 2024), and the results were reported as a score plot. The location taken up by each sample in the score plot of PC1 (explaining 52.3% of the variance) vs. PC2 (explaining 29.9% of the variance) is shown in Appendix A, while those in the score plot of PC1 vs. PC3 (explaining 13.9% of the variance) are displayed in Appendix A. As is observable both in Appendix A, the PC1 samples were separated based on their crosslinking percentage and degradation profiles. Non-crosslinked Gel B, which has demonstrated the highest degradability, was positioned on the right side of the plot, with a high positive score, while the reticulated samples were all grouped on the left side, at scores close to zero or negative. On the contrary, the mass loss percentage and/or degradation rate improved when moving from the left side of the score plot to the right one, with M6 and M9 being the samples with the lowest mass loss. M1, M2, M4, and M5, demonstrating similar mass losses, were located all at scores close to zero for PC1. These results evidenced that the degradation of all samples was strictly correlated with their reticulation percentage. Additionally, in the plot in Appendix A can be observed a separation of the samples for PC2 as well, mainly based on their kinetics of degradation. While all samples whose degradation followed PSO kinetics were positioned at scores close to zero or negative, M3 was located at a higher positive score, thus confirming its unique degradative behavior, which fit Korsmeyer–peppas kinetics.

#### 2.1.9. Equilibrium Water Loss Rate (%)

The water loss profiles of the crosslinked gelatines were obtained by heating the swollen samples over time and measuring their weight loss until no variation in their mass was detected. This point indicated the time necessary for the systems to reach their equilibrium water loss (*WL e*). Such experiments were performed to evaluate the capability of the prepared swollen scaffolds to retain water when air-exposed. All experiments were performed in triplicate, and the results were expressed as means ± standard deviation (SD).

By plotting the cumulative water loss percentages (%) of all samples vs. time, the graphs shown in Figure 9 were obtained. 

According to Figure 9, samples M4, M5, M6, and M9 fully released the water they previously absorbed and a *WL e* of 100% was reached after 22 h 15′ by M4 and after 23 h 51′ by M5, M6, and M9. On the other hand, the *WL e* values of M1, 2, and 3 were 92%, 90%, and 95%, respectively, after 22 h 15′, thus establishing that M2 was the sample more efficient at retaining water (10%). Kinetics studies, whose results are available in Appendix A and in Appendix A, established that the water losses from our samples were governed by different mechanisms. Particularly, the water loss from M1, M2, and M9 was governed by zero-order kinetics, that from M4 by first-order kinetics, that from M5 and M6 by Korsmeyer–peppas kinetics, and that from M3 by Hixson–Crowel kinetics. 

The experimental data reported in Figure 9 were modelled with zero-order kinetics using the plot of the cumulative water loss (%) vs. time and generating the linear regressions of the obtained dispersion graphs to evaluate their R^2^ [99]. Zero-order kinetics are described by Equation (7).
(7)(WL%)t=(WL%)o+Ko×t
where *(WL%)t* is the cumulative water loss percentage at time *t*, *(WL%) o* is the cumulative water loss percentage at the beginning (*t = to*), and *Ko* is the zero-order constant. Accordingly, *Ko* corresponded to the slopes of the light blue equation of M1, the fuchsia equation of M2, and the purple equation of M9 in Appendix A, while *(WL%) o* was their intercepts. To model the experimental data using first-order kinetics, the Log of their cumulative water loss (%) was plotted vs. time [99]. First-order kinetics are described by Equation (8)
(8)$$Log\left(WL\%\right)t=Log\left(WL\%\right)o+\frac{K1}{2.303}\times t\;$$
where (*WL*%) *t* is the cumulative water loss percentage at time *t*, *(WL%) o* is the cumulative water loss percentage at the beginning (*t = to*), and *K*1 is the first-order constant. Accordingly, *K*1/2.303 corresponds to the slope of the yellow equation of M4 in Appendix A and *Log* (*WL*%) *o* was its intercept. When the experimental data were modelled with Korsmeyer–peppas kinetics, the Ln cumulative water loss (%) was plotted vs. Ln time. Korsmeyer–peppas kinetics are described by Equation (5), which was previously reported. Accordingly, Ln *K_KP_* corresponded to the intercepts of the red equation of M5 and green equation of M6 in Appendix A, and *n* was their slopes. Finally, to model the experimental data with Hixson–Crowell kinetics, the cubic roots of the water residual (%) were plotted vs. time. Hixson–Crowell kinetics are described by Equation (9).
(9)$${\;}^{3}\sqrt{1-(F\%)t}=1-(KHC)t\;$$
where (*F*%) *t* is the cumulative fraction percentage of water loss on time *t* and *K*_HC_ is the Hixson–Crowell constant, corresponding to the slope of the fuchsia equation of M3 in Appendix A. The obtained values of *K* for all samples, according to their best-fitting model, have been included in Table 6. The *n* values reported in Table 6 for M5 and M6, very close to 0.5, establish that the main mechanism governing their water loss was a Fickian-type diffusion from thin films. 

#### 2.1.10. Rheological Studies

Rheology is a fundamental parameter to be considered when scaffolds for cell adhesion and proliferation studies are prepared. From the data collected by performing rheological studies on the scaffolds M1–M6 and M9, hydrated up to their *Q e*, two types of graphs were constructed: one reporting the shear stress (*τ* [mPa]) as a function of the shear rate (*γ* [s^−1^]) (Figure 10a) and another reporting the viscosity (*η* [mPa × s]) as a function of *γ* (Figure 10b).

Figure 10a shows that τ was not directly proportional to *γ* for all samples under study, thus meaning that viscosity (*η*) was not constant but decreased dramatically with small increases in *γ* (Figure 10b). These results demonstrated that all samples behaved as shear thinning fluids, differently from so-called shear thickening or dilatant ones, whose viscosity increases with intensifications of *γ*. However, for values of *γ* > 50–60 1/s, *η* was practically constant, and it did not change significantly for greater increases in *γ* (Figure 10b). Collectively, all our samples demonstrated a pseudoplastic non-Newtonian behavior, not respecting the Newtonian law expressed by Equation (10) for *γ* < 50 1/s.
*τ* = *γ* × *η*(10)

Otherwise, all our samples behaved as Bingham-type fluids for *γ* over 50–60 1/s and for τ > the values given by the intercepts of their linear tracts with the y axis, namely the yield stress. Fluids characterized by such behavior are defined as Bingham pseudoplastic materials. 

The data on the shear stress vs. shear rate of these fluids can be modeled using the Herschel–Bulkley viscosity model, which provides the index flow (*n*), useful for confirming the shear thinning behavior of our samples. 

The Herschel–Bulkley rheological model is expressed by Equation (11).

*τ* = *τ_oH_* + *K_H_* × *γ* × *n _H_*(11)

where *n_H_* is the Herschel–Bulkley flow behavior index, which indicates the tendency of a fluid to shear thin or thick; *K_H_* is the consistency coefficient, which serves as the viscosity index of the systems; and *τ_oH_* is the Herschel–Bulkley yield stress point, over which *η* becomes constant.

According to the literature [81], the Hershel–Buckley model can be fitted to data on shear stress and shear rate by reporting it in a graph of *Log (τ−τ_o_)* vs. *Log (γ)*, where *τ_o_* is the value of *τ* corresponding to the minimum *γ* applied. Appendix A shows the dispersion graphs obtained for all samples, the equations of their linear regression models, and their coefficients of determination (R^2^), whose values are parameters useful for judging if the mathematical model fit experimental data well.

For each sample, *Log K* and *n* were obtained from the equations in Appendix A, where *Log K* was the intercept and *n* the slope. The *K* value was calculated accordingly and inserted in Table 7, together with the equations of the linear regressions obtained by the Herschel–Buckley mathematical model, the coefficients of determination (R^2^), the values of the slope and intercept, and the values of Herschel–Buckley yield stress. 

Particularly, when *n* < 1, the fluid is shear-thinning (a pseudoplastic fluid); when *n* = 1, the fluid is Newtonian; and when *n* > 1, the fluid is shear-thickening (dilating hydrogels). Accordingly, the shear-thinning behavior previously assumed for M1–M6 and M9 was confirmed. Analogously, the data on the viscosity vs. shear rate of these fluids can be modeled using two different forms of the Cross rheological model, which are expressed by Equations (12) and (13).
*Log* *η* = *Log* (*η_o_*/*α*) − *n Log γ*(12)
*η* = *η*∞ + (*η*_*o*_/*α*) × *γ*^−*n*(13)

In these equations, *γ* is the shear rate; *η_o_* is the viscosity when the shear rate is close to zero; *η∞* is the viscosity when the shear rate is infinity; *n* is the flow behavior index, which indicates the tendency of a fluid to shear thin or thick; and *α* is the consistency index, which serves as the viscosity index of the systems. According to the literature, two Cross models can be obtained by graphically reporting *Log η* vs. *Log γ* (Appendix A) and *η* vs. *γ^−n* (Appendix A) [86,103]. Both Appendix A show the dispersion graphs obtained for all samples, the equations of their linear regression models, and their related coefficients of determination, which are parameters useful for judging if the mathematical models fit the experimental data well. For each sample, *Log (η_o_/α)* and n were obtained from the equations in Appendix A, where *Log (η_o_/α)* was the intercept and n the slope. Additionally, as previously reported, when *n* < 1, the material is a shear-thinning pseudoplastic fluid; when *n* = 1, the fluid is defined as Newtonian; and when *n* > 1, the fluid behaves as a shear-thickening dilating hydrogel. Then, for each sample, *η∞* and *η_o_/α* were obtained from the equations in Appendix A, where *η∞* was the intercept and *η_o_/α* the slope. According to the literature, *η∞* served to determine the viscosity of samples at a very low shear rate (*η_o_*) using the relationship *η _o_* = 1000 × *η∞* [85], and the obtained values of *η_o_* were used to determine the *α* values, using either the intercepts of the equations in Appendix A or the slopes of the equations in Appendix A. All results obtained have been included in Appendix A. The *α* values obtained from fitting experimental data with the Cross model expressed in Equation (12) were sufficiently in agreement with those obtained by modeling data with the Cross Equation (13). From the values of R^2^, we can demonstrate that the Cross rheological equations fit the experimental data of *η* vs. *γ* better than the Herschel–Buckley one, when used to model the experimental data of *τ* vs. *γ*. In this regard, while samples M2–M6 demonstrated a shear thinning Bigham pseudoplastic behavior, as in the previous case, M1 and M9 provided *n* = 1, evidencing that M1 and M9 behave as pseudoplastic fluids only at very low values of shear rate and are, in general, more similar to Bingham-type materials (*η* constant, but *τ_o_* ≠ 0).

##### PCA of Rheological Data 

The rheological data of all samples were further processed using a PCA, as previously performed for spectral data, as well as for crosslinking and mass loss. 

This time, we collected the data on viscosity and shear stress and arranged them together with all parameters calculated by the Herschel–Buckley and Cross equations in a 7 × 40 matrix (280 variables). The results were presented as a score plot of PC2 (27.6% of the variance) vs. PC1 (58.4% of the variance). In this case, the PCA allowed us to visualize the reciprocal positions that our samples occupied, depending on the presence of similarities or differences in their rheological characteristics (Figure 11).

The samples were well separated both on PC1 and PC2. From an examination of the score plot, it can be assumed that, on PC1, samples were located based on their values of *η_o_* and *α*, expressed as *Log (η_o_/α)* (the intercept of Equation (12)) and *η_o_/α* (the slope of Equation (13)), with the samples with higher values positioned at negative scores, and those with lower values at positive scores. On the other hand, on PC2, samples were clustered based on their rheological behavior and n values. Particularly, M1 and M9, with n values close to 1 (Bigham behavior), were located close to each other and at negative scores, while other the samples, all with 0 < *n* >1, were on the right side of the plot. 

## 3. Materials and Methods

### 3.1. Chemicals and Instruments

All reagents and solvents were from Merck (formerly Sigma-Aldrich, Darmstadt, Germany). The solvents were dried and distilled according to standard procedures. Attenuated total reflectance (ATR) Fourier transform infrared (FTIR) spectra were acquired on the same instrument as previously reported [104]. UV spectra were recorded on an HP 8453 UV-visible System spectrophotometer (Agilent, Santa Clara, CA, USA) using quartz cuvettes with an optical path of 1 cm. Potentiometric titrations were performed with a Hanna Micro-processor Bench pH Meter (Hanna Instruments Italia srl, Ronchi di Villafranca Padovana, Padova, Italy). For the rheological characterization of the scaffolds, a concentric cylinder viscometer (Phisica Haake Thermo Fisher Scientific Inc., Waltham, MA, USA) equipped with a Z4 probe was used. 

### 3.2. Gelatine B (Gel B) Crosslinking 

The copolymers used in this study to reticulate gelatine B (Gel B) (Figure 1) were prepared as previously described [8,62,63]. The protocol recently developed by us to crosslink gelatine by means of amine- or aldehyde-containing synthetic copolymers with or without amine oxidases (LO or PAO) has been recently described [8]. Briefly, about 500 mg of Gel B (Merck, Darmstadt, Germany) and the appropriate buffer system were introduced into a 50 mL container (Ø ext 35 mm, Ø int 28 mm, h = 10 cm) under heating at 40 °C, until the Gel B was completely dissolved (about 30 min). Solutions in the same buffer system of the appropriate copolymer and of LO or PAO were added to the Gel B solution. The mixture was left under heating at 40 °C for 24 h. After this time, the reaction mixture was subjected to dialysis. To this end, before the end of the reaction, the dialysis membrane was placed, to condition, in approximately 100 mL of mQ H_2_O. One side of the membrane was then fixed with a special clip and the contents of the test tube were transferred into the membrane, whose upper side was closed as well. The membrane was immersed in mQ H_2_O and left stirring for 24 h. The dialyzed solution was transferred to a 500 mL beaker and the membrane was washed with mQ H_2_O. Then, all the dialyzed material and washings were frozen at −18 °C for 24 h and then lyophilized with a freeze-dryer Buchi Lyovapor L200I S (Büchi Labortechnik AG, Flawil, Switzerland). The fully dried crosslinked gelatines were then extracted with water at 40 °C or DMSO at 70 °C, in the case of the CPMA/DMAA samples, for 24 h. The residual aqueous solvent was removed by lyophilization, as previously described, obtaining the crosslinked gelatines in the form of fluffy solids. The data and results concerning each experiment carried out in this study are available in Table 1 (Section 2.1.2).

### 3.3. UV Titrations of Crosslinked Gelatines

A sample of crosslinked gelatine (approximately 11 mg), 1 mL of a 4% NaHCO_3_ solution, and 1 mL of a 0.5% TNBS (2,4,6-trinitrobenzenesulfonic acid) solution were introduced into a 50 mL test tube. The thus-obtained orange suspension was heated in an oil bath at 40 °C for 4 h. Then, 3 mL of a 6 M HCl solution was added, with effervescence and the appearance of a yellow coloration observed. The mixture was then heated to 116 °C for 1 h. The solution was transferred into a 100 mL separating funnel, the test tube was washed with 5 mL of mQ H_2_O and extracted with Et_2_O (3 × 4 mL). The aqueous phase was heated in a water bath for 15 min to eliminate traces of residual ether. A total of 5 mL of aqueous phase was then taken with a single-notched pipette and diluted to 20 mL in a flask, using mQ H_2_O. A solution sample prepared in the same conditions but using pristine Gel B (11.2 mg) was used as a blank. UV analysis was performed at 346 nm. The results of the titrations are available in Table 2 (Section 2.1.3).

### 3.4. NaOH Titration of Crosslinked Gelatines

About 220–250 mg of crosslinked gelatine and 15 mL of mQ H_2_O were introduced into a 50 mL beaker equipped with a magnetic stirrer and left stirring for 25 min, thus allowing the gelatine to swell and making the titratable groups better accessible. The titration was carried out with a standard solution of NaOH 0.1033 N, adding aliquots of 0.06 mL and measuring pH values. The experiments were carried out in triplicate and the results were expressed as the mean of three independent measures ± SD. Titration data were reported in a graph (pH vs. NaOH (mL), obtaining titration curves, from which the volume of NaOH at pH values equal to 8.50 and 11.50 were extrapolated. The results of the titrations are available in Table 2 (Section 2.1.3).

### 3.5. ATR-FTIR Spectra

ATR-FTIR analyses were carried out on Gel B, CP5/DMAA, and CPMA/DMAA copolymers, as well as on crosslinked gelatines M1–M6 and M9, directly on the solid samples. The spectra were acquired from 4000 to 600 cm^−1^, with a 1 cm^−1^ spectral resolution, co-adding 32 interferograms, and with a measurement accuracy in the frequency data at each measured point of 0.01 cm^−1^ due to the internal laser reference of the instrument. Acquisitions were made in triplicate, and the spectra shown in Section 2.1.4 are the most representative images. 

### 3.6. Optical Microscopy Analyses

The morphology of samples M1–M6 and M9 were here investigated via an optical microscopy (OM) analysis. In the performed experiments, dried samples of crosslinked gelatines were observed using a Leica DM750 optical microscope (Leica Italy, Milan, Italy) equipped with 4×, 10×, and 40× objectives. Sequential images were acquired with the 4×, 10×, or the 40× objectives. The camera used for image capture was a Leica ICC50W (Leica Italy, Milan, Italy). All images were processed by means of LAS EZ 3.4.0. software (Leica Italy, Milan, Italy).

### 3.7. Equilibrium Swelling Rate

The swelling measurements were carried out at room temperature by immersing 10.0–46.0 mg of fully dried crosslinked gelatine in deionized water (pH = 7.2–7.3) in a test tube [81,105]. At intervals of time selected according to the literature [81,82], weight measurements were made until the weight was constant. The cumulative swelling ratio percentage (*Q %*), as function of time, was calculated from Equation (14).
(14)$$Q\;\left(\%\right)=\frac{Ws\;(t)-Wd}{Wd}\times 100$$
where *Wd* and *Ws (t)* were the weights of the fully dried and swollen crosslinked gelatine at time *t*, respectively. The equilibrium swelling rate (*Q e*) was determined at the time point (time *t*) the hydrated gelatine achieved a constant weight, which should correspond to the water absorption capacity (WAC) of the sample.

### 3.8. Water Absorbing Capacity (WAC %)

As previously described [81,105], for determining the water absorbing capacity of crosslinked gelatines, samples of fully dried crosslinked gelatines were placed in a graduated centrifuge tube (Ø ext = 14 mm) and the volume (mL) of each sample was determined (*Vi*). An excess of water was added (10 mL) and left at room temperature overnight. Upon centrifugation at 6000 rpm for 10 min, and if necessary for an additional 20 min at 4000 rpm, the volumes reached by the swollen gelatines was measured (*Vf*). The values of *Vi* and *Vf* were used to calculate the water absorbing capacity percentage (*WAC %*) according to the following Equation (16):(15)$$WAC\;\left(\%\right)=\frac{Vf-Vi}{Vi}\times 100$$
where *Vi* and *Vf* are the volume of the dried crosslinked gelatine and that of the swollen material, respectively [81].

### 3.9. Porosity 

To determine the porosity of the crosslinked gelatines, intended as their equilibrium water content (EWC), the solvent replacement method was used. We proceeded as in the experiment described in the previous (Section 3.8) and the porosity was calculated from the following Equation (17) [81,105]:(16)$$Porosity\;\left(\%\right)=\frac{V2-V1}{\rho V2}\times 100$$
where *V*1 is volume of the crosslinked gelatine before its immersion in water, *V*2 is the volume of the hydrogel after its immersion in water, and *ρ* is density of water (*ρ* = 1). 

Porosity was also determined by a gravimetric method, calculating first the relative density of each sample according to the formula *ρ*_R_ = *ρ**/*ρ*
_G_, where *ρ** is the apparent density of the fully dried aerogels, while *ρ*
_G_ is the dry density of solid gelatine, assumed to be 1.369 g/mL [106]. The apparent density (*ρ**) was in turn calculated according to the formula *ρ** = m/V, where m is the mass and V is the volume of the crosslinked gelatines [91]. Collectively, porosity, defined as the percentage of void space in a 3D sponge, was determined from these relative density values according to Equation (18) [89,106,107]:(17)$$Porosity\;\left(\%\right)=(1-\frac{\rho *}{\rho \;G})\times 100$$

### 3.10. Biodegradability of Crosslinked Gelatins over Time by In Vitro Mass Loss Experiments

Samples of vacuum-dried crosslinked gelatine were inserted in centrifuge tubes, with 10 mL of PBS added, and incubated at 37 °C. At fixed time points of 4, 8, 14, and 21 h, and then 1, 2, 3, 4, 6, and 8 days, the formed hydrogels in the tubes were centrifugated to remove the supernatant PBS, and the water on the hydrogels’ surface was wiped off by inverting the tubes. The hydrogels were then dried and weighed to record their mass change with time. The weight of the hydrogels at the pre-set time intervals has been indicated as *Mt*. The cumulative mass loss (*ML*) percentages (mass change) over time were calculated according to the following Equation (15):(18)$$Mass\;Loss\;\left(ML\right)\left(\%\right)=\frac{Mi-Mt}{Mi}\times 100$$

In (15), *Mi* and *Mt* are the initial mass of the dry crosslinked gelatine and the mass of remaining polymer after time *t* in PBS, respectively.

### 3.11. Equilibrium Water Loss Rate

Samples of fully swollen crosslinked gelatines in test tubes, with a water content in the range of 11.0–325.0 mg, were placed in an oven at 37 °C. The water loss was checked over time until the weight of the samples was constant. Their cumulative water loss percentages were determined by Equation (19):(19)$$Water\;Loss\;\left(\%\right)=\frac{Wi-Wt}{Wi}\times 100$$

### 3.12. Rheological Properties of Crosslinked Gelatines

For the rheological characterization of the crosslinked gelatines, a concentric cylinder viscometer (Phisica Haake Thermo Fisher Scientific Inc., Waltham, MA, USA) equipped with a Z4 probe was used. For the determination of dynamic viscosity and shear stress, approximately 5 mL of the crosslinked gelatines, soaked to their equilibrium degree of swelling (EDS), was placed in the viscometer and thermally equilibrated at the test temperature (25 ± 0.1 °C) for 60 min before the measurement. Then, the samples were subjected to an increasing shear rate from 1 to 100 s^−1^. All the experiments were performed in triplicate and the results were expressed as mean ± SD.

## 4. Conclusions

The main scope of this study was to develop and characterize new scaffolds and detect those possessing physicochemical properties promising for applications in tissue engineering (TE). To this end, a protocol recently developed by us for the enzymatic crosslinking of gelatine (Gel B) without resorting to commonly used crosslinking agents, has been applied, achieving nine gelatine-based aerogel-like composites, of which seven (M1–M6 and M9) demonstrated yields (%) and crosslinking degrees that made them suitable for further physicochemical characterization. Following several experiments, the chemical structure, the morphology, the equilibrium swelling rate (%), the water absorption capacity (%), the apparent density and porosity (%), the equilibrium water content (%), the biodegradability, the water loss rate (%), and the rheological properties of all these aerogel-like composites were investigated. While M3 showed the best degradation profile, M4, M5, M6, and M9 demonstrated the lowest apparent density, the highest swelling speed and capacity, and the highest porosity (%), which were like those of previously reported materials described as being successful in in vitro and in vivo biological experiments. Moreover, M4 and M6 demonstrated a crater-like morphology with cavities with an average size of 18 µm, M5 showed a singular labyrinth-like morphology with smaller cavities of 16 µm, while M9 showed cavities of 93 µm. From these results, and according to the literature, M3, M4, M5, M6, and M9 can be considered very promising materials for supporting several cellular activities such as cell proliferation, osteogenesis, adipogenesis, cell infiltration, and adult mammal skin cell regeneration, whose proliferation is reported to be promoted by scaffolds with pores of 10–125 µm. For these, M3, M4, M5, and M6, produced using the CP5/DMMA copolymer, which has been reported to be non-cytotoxic, could be the most biocompatible choice. Further studies are in progress to determine the effective interactions of the selected scaffolds with cells.

## Data Availability

The data supporting the reported results are included in this manuscript and in the related Appendix A.

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
