# Peer review of "Synthesis and Physicochemical Characterization of Gelatine-Based Biodegradable Aerogel-like Composites as Possible Scaffolds for Regenerative Medicine"

_ijms, 2024, doi:10.3390/ijms25095009_

Round 1

Reviewer 1 Report

Comments and Suggestions for Authors

The hydrogels, such as gelatin-based ones, has attracted huge interests due to their unique properties, such as biocompatibility, biodegradability, suitable mechanical properties, and mimicking extracellular matrix. The development of new approaches and protocols for the synthesis of 3D gelatin-based constructions is of fundamental importance. The authors of the study have developed 9 different protocols to form gelatin-based aerogels composites with a variety of the morphology, mechanical and biodegradation properties. This work is promising and provides new data on the synthesis of these materials. However, there are several important issues, which should be carefully considered for a major revision of the work.

Major comments:

 1.       The abstract’s content should be significantly improved. Now, the abstract consists of the “dry” listing of done work (experiments, samples name, etc.) without the focused description of the observed results, changes, and trends between different samples, which highlight a novelty of the work. This will provide a strong feeling of the fundamental important of the work for the readers. Moreover, authors used a phrase “obtaining nine samples”, which can be considered as a total number of samples in the study or different groups of samples. So, it can cause misunderstanding for readers. Thus, the abstract should be carefully revised and restructured.

 2.       The additional characterization of the morphology and topography is highly required using common and precise techniques, such as at least SEM, and may be additionally AFM techniques to see topography changes in micro and nano-ranges. The optical microscopy depends on and is strictly limited by the focus distance. Moreover, light can be scattered differently in areas of different densities and geometries. Therefore, the optical microscopy cannot be considered as a precise method to evaluate the morphological and topographical changes, such as a pore size and cavities, caused by different crosslinking protocols.

 3.       The chemical composition and structure characterization is highly required more comprehensive approach to show more detailed changes in quantitative and qualitive manners between different groups of samples. ATR-FTIR spectra is very sensitive techniques, which is limited by well-known several drawbacks, such as humidity surrounding environment and samples surface, roughness of the samples. Moreover, it is challenging to provide some numerical data about changes in the structure of the different composites. Due to this, a more comprehensive approach is required. For this purpose, I would strongly advice to use XPS and/or Raman spectroscopies, which both have many advantages over ATR-FTIR. The combination of several techniques will provide better understanding of the caused changes in the composition and structure of the hydrogels.

Minor comments:

 1.       There is no scale bar on Figure 3d.

2.       The scale bar of Figure 4D is significantly different from other images in this figure. If the objective was the same for all composites, then all images should have the same scale bar.

3.       It is necessary to use the scientific notation of values on the Y scale in Figure 10.

4.       All tables in the manuscript should have the same style.

5.       The quality of the Figure 11 should be improved, including the font size in Figure should be increased, improving readability.

Author Response

The hydrogels, such as gelatin-based ones, has attracted huge interests due to their unique properties, such as biocompatibility, biodegradability, suitable mechanical properties, and mimicking extracellular matrix. The development of new approaches and protocols for the synthesis of 3D gelatin-based constructions is of fundamental importance. The authors of the study have developed 9 different protocols to form gelatin-based aerogels composites with a variety of the morphology, mechanical and biodegradation properties. This work is promising and provides new data on the synthesis of these materials. However, there are several important issues, which should be carefully considered for a major revision of the work.

We thank the Reviewer for having considered the topic of our work of fundamental importance and data provided original. Anyway, we kindly point out that we have not developed 9 protocols but a versatile one. Its adaptability has allowed us to use it to prepare nine different samples using different conditions.

Major comments:

  1. The abstract’s content should be significantly improved. Now, the abstract consists of the “dry” listing of done work (experiments, samples name, etc.) without the focused description of the observed results, changes, and trends between different samples, which highlight a novelty of the work. This will provide a strong feeling of the fundamental important of the work for the readers. Moreover, authors used a phrase “obtaining nine samples”, which can be considered as a total number of samples in the study or different groups of samples. So, it can cause misunderstanding for readers. Thus, the abstract should be carefully revised and restructured.

We thank a lot the Reviewer for his/her precious comments. In this regard, the abstract has been fully reformulated following the Reviewer’s suggestions. Concerning the expression signaled as misleading by the Reviewer, it has been modified. Anyway, as above specified, the number nine refers to the total number of prepared samples and not to groups of samples or to the number of protocols developed in our previous work.

  1. The additional characterization of the morphology and topography is highly required using common and precise techniques, such as at least SEM, and may be additionally AFM techniques to see topography changes in micro and nano-ranges. The optical microscopy depends on and is strictly limited by the focus distance. Moreover, light can be scattered differently in areas of different densities and geometries. Therefore, the optical microscopy cannot be considered as a precise method to evaluate the morphological and topographical changes, such as a pore size and cavities, caused by different crosslinking protocols.

As above reported, we did not use different protocols to reticulate Gel B, but the same versatile protocol developed in our previous work (Ref 8, revised version) in different condition of reactions (see Table 1) achieving nine different crosslinking compounds (M1-M9), subsequently fully characterized. Concerning the suggestion of the Reviewer, we agree with him/her about the higher precision of SEM and AFM techniques to characterize morphology and topography of micro and nanomaterials and to evaluate possible changes depending on different crosslinking conditions. Anyway, to reduce the studies costs, before recovering to these techniques not available in our laboratory, we have tried to observe our materials with our optic microscope as reported also in literature (https://doi.org/10.1016/j.jmrt.2020.12.006 https://doi.org/10.1038/s41598-018-20438-6), hoping to succeed in detecting acceptable images evidencing differences and measurable cavities. The results, in our opinion, are satisfactory. So, we kindly ask the Reviewer to not force us to carry out further analyses, impossible to be performed within 10 days and too much expensive for us, and to accept the optical microscopy used by us. Specification about this question have been inserted in the main text. Please, see lines 163-169.

The chemical composition and structure characterization is highly required more comprehensive approach to show more detailed changes in quantitative and qualitive manners between different groups of samples. ATR-FTIR spectra is very sensitive techniques, which is limited by well-known several drawbacks, such as humidity surrounding environment and samples surface, roughness of the samples. Moreover, it is challenging to provide some numerical data about changes in the structure of the different composites. Due to this, a more comprehensive approach is required. For this purpose, I would strongly advice to use XPS and/or Raman spectroscopies, which both have many advantages over ATR-FTIR. The combination of several techniques will provide better understanding of the caused changes in the composition and structure of the hydrogels.

As specified in the replay to point 1, the number nine refers to the total number of prepared samples and not to groups of samples. However, we agree with the Reviewer that combination of several techniques will provide better understanding of the caused changes in the composition and structure of the hydrogels, but we have already characterized our samples by a very high number of techniques and experiments, that further ones would only burden the work, without adding more useful information. ATR-FTIR spectroscopy is a fast, operator-friendly, and conservative technique, stronger that Raman (please, see this https://www.mt.com/us/en/home/applications/L1_AutoChem_Applications/Raman-Spectroscopy/raman-vs-ir-spectroscopy.html), with a plethora of qualitative and quantitative applications in several sectors, also in industry where it is often exploited also in quality control. Specifically, it relies on the absorption of light by molecules at frequencies corresponding to the fundamental vibrations of those molecules. Molecules with functional groups that have strong dipoles display strong peaks in the IR, whereas functional groups that have weak dipoles and readily undergo a change in polarizability display strong peaks in Raman. Unfortunately, not all molecules are Raman active and for some of those that are Raman active, they may fluorescence in the presence of NIR and/or visible laser frequencies. Fluorescence is especially problematic with Raman because it is orders of magnitude stronger in signal than Raman scattering and often results in overwhelming the Raman signal. Additionally, although when used in solution ATR-FTIR analyses could be exploited for quantitative scopes, here, as in several published papers reporting on composite materials, including aerogels, the ATR-FTIR or the more conventional FTIR spectroscopy were not used to obtain numerical data (as indicated by the Reviewer as a limitation of ATR-FTIR), but only to get qualitative information about the functional groups present in the composites. Anyway, for more clarity, explanations on the reasons of using this technique rather than those suggested by the Reviewer have been included in the text (lines 162-163 and 245-248) Furthermore, we make kindly note to the Reviewer, that in our work, to obtain the most significant qualitative information form the FTIR analyses, the obtained spectral data were further processed by PCA with interesting conclusions. As us, several other authors have used FTIR analyses in place of Raman or XPS to characterize their aerogels. The following article is an illuminating case https://doi.org/10.1016/j.molstruc.2008.08.025. Others: https://doi.org/10.1016/j.ceramint.2024.02.272, https://doi.org/10.1016/j.jobe.2023.108243, https://doi.org/10.1016/j.jnoncrysol.2024.122859, https://doi.org/10.1016/j.jnoncrysol.2021.121048, 10.1007/s10853-018-2553-4, https://link.springer.com/article/10.1007/s00604-020-04443-z. These references have been included in the main text.

Minor comments:

  1. There is no scale bar on Figure 3d.

As required the scale bar has been inserted in Figure 3d.

  1. The scale bar of Figure 4D is significantly different from other images in this figure. If the objective was the same for all composites, then all images should have the same scale bar.

We thank a lot the Reviewer for his/her attention. The Figure caption was wrong. Now it has been corrected (lines 310-311).

  1. It is necessary to use the scientific notation of values on the Y scale in Figure 10.

The graph has been provided as such by the Microsoft Excel software. Graphs in other works by us with the same notation have not been questioned both by Reviewers and by IJMS. We kindly ask the Reviewer to accept the present notation.  

  1. All tables in the manuscript should have the same style.

As required, all Tables have been uniformed and now have the same style.

  1. The quality of the Figure 11 should be improved, including the font size in Figure should be increased, improving readability.

As required, Figure 11 has been improved and now it is clearly readable.

Reviewer 2 Report

Comments and Suggestions for Authors

The article underscores the ongoing and significant investigations of the authors on the synthesis and characterization of gelatine based scaffolds for regenerative medicine. This research builds upon a recently published paper in the Int. J. Mol. Sci. 2024, 25, 2897; Sci. 2024, 25, 2897; https://doi.org/10.3390/

Major suggestions:

Rows 36-37: “appeared more recently in a paper dated to 1993 [3]”. Recently" typically refers to a period of time that is not long ago. Improvement is needed.

Consider improving the language in row 53. The term' defective patients' is inappropriate. You could use 'patients requiring intervention' or another suitable phrase.

Row 57: Figure S1 is the same as Figure 1, which was previously published in the paper cited above. Remove it from the supplementary material and address it as a citation.

There is a big similarity between the introductions of the recent manuscript and the paper mentioned above. Examples:

Rows 58-60: Being the number of patients requiring transplantation enormous, compared to that of available organs, research in TE is of paramount relevance.

Published paper: The application potential of TE is enormous due to the great disproportion existing on a global scale between the number of patients requiring transplantation and the number of organs available.

Rows 60-61: TE provides the possibility to have effective substitutes for liver, kidneys, pancreas, as well as for tissues such as blood vessels, skin, cartilage, bones, ligaments, and tendons.

Published paper: Moreover, enormous is the variety of organs such as the liver, kidneys, and pancreas, or tissues such as blood vessels, skin, cartilage, bones, ligaments, and tendons, for which it would be very useful to have effective substitutes

Ðœany other similarities have been found within the Introduction section. I strongly recommend the Introduction be re-written to clearly indicate the advantages and necessity of the recent work.

Row 127: Scheme S1, Supplementary Materials has already been published (in the paper mentioned above). To be addressed only as a citation and removed from the supplementary.

Row 135: Scheme S2, Supplementary Materials has already been published (in a paper mentioned above). To be addressed only as a citation. To be removed from the supplementary.

All codes and abbreviations used must be explained at first appearance. For example, “11b, rows 131, 172” is explained in row 187. Please correct.

Please provide NMR spectra of newly synthesized composited, co-polymers and intermediates.

 The authors stated that “ Due to these characteristics that well fit within the requirements for cell adhesion, infiltration, and proliferation, aerogel-like composites M4, M5, M6 and M9 could be suitable candidates for applications in TE and regenerative medicine.” No in vivo cell experiments were performed to prove it. Please provide the necessary set of experiments – cell viability, proliferation, differentiation, etc.

There is more than 15% self-citation (53, 54, 55, 56, 63, 67, 68, 77, 82, 87, 88, 89, 90, 91, 92). Please consider decreasing it.

Author Response

The article underscores the ongoing and significant investigations of the authors on the synthesis and characterization of gelatine based scaffolds for regenerative medicine. This research builds upon a recently published paper in the Int. J. Mol. Sci. 2024, 25, 2897; Sci. 2024, 25, 2897; https://doi.org/10.3390/

Major suggestions:

Rows 36-37: “appeared more recently in a paper dated to 1993 [3]”. Recently" typically refers to a period of time that is not long ago. Improvement is needed.

As required by the Reviewer, the sentence has been improved, by replacing “more recently” with “later” (line 48).

Consider improving the language in row 53. The term' defective patients' is inappropriate. You could use 'patients requiring intervention' or another suitable phrase.

We thank a lot the Reviewer for his/her suggestion, which has been applied in line 64.

Row 57: Figure S1 is the same as Figure 1, which was previously published in the paper cited above. Remove it from the supplementary material and address it as a citation.

As asked, Figure S1 in Supplementary Materials has been removed and the main text has been modified to insert the simple citation as suggested by the Reviewer. Please, see lines 68-69.

There is a big similarity between the introductions of the recent manuscript and the paper mentioned above. Examples:

Rows 58-60: Being the number of patients requiring transplantation enormous, compared to that of available organs, research in TE is of paramount relevance.

Published paper: The application potential of TE is enormous due to the great disproportion existing on a global scale between the number of patients requiring transplantation and the number of organs available.

Rows 60-61: TE provides the possibility to have effective substitutes for liver, kidneys, pancreas, as well as for tissues such as blood vessels, skin, cartilage, bones, ligaments, and tendons.

Published paper: Moreover, enormous is the variety of organs such as the liver, kidneys, and pancreas, or tissues such as blood vessels, skin, cartilage, bones, ligaments, and tendons, for which it would be very useful to have effective substitutes…

Ðœany other similarities have been found within the Introduction section. I strongly recommend the Introduction be re-written to clearly indicate the advantages and necessity of the recent work.

We thank the Reviewer for his/her suggestions. Sentences duplicate from our previous work have been removed or modified. Some parts along the Introduction have been reformulated to better indicate the advantages and necessity of this recent paper. Please, consider the several changes made to the Introduction section.

Row 127: Scheme S1, Supplementary Materials has already been published (in the paper mentioned above). To be addressed only as a citation and removed from the supplementary.

As asked, Scheme S1 in Supplementary Materials has been removed the main text has been modified to insert the simple citation as suggested by the Reviewer. Please, see lines 140-145.

Row 135: Scheme S2, Supplementary Materials has already been published (in a paper mentioned above). To be addressed only as a citation. To be removed from the supplementary.

As asked, Scheme S2 in Supplementary Materials has been removed and the main text has been modified to insert the simple citation as suggested by the Reviewer. The same for Scheme S3. Please, see line 145-147.

All codes and abbreviations used must be explained at first appearance. For example, “11b, rows 131, 172” is explained in row 187. Please correct.

The explanation required by the Reviewer has been included in the caption of Figure 1 mentioned in the text at the first mention of CP5, CP11b and CP11c. Please, see lines 153-155.

Please provide NMR spectra of newly synthesized composited, co-polymers and intermediates.

There are no new copolymers or intermediates. The copolymers used in this study to crosslink Gel B and the monomers used to prepare them have been completely characterized in our previous work (Ref 8, revised version), cited by the Reviewer more than one time, where all FTIR and NMR spectra have been given. Overwise, the crosslinked materials, since reticulated, are insoluble in all organic solvents and swell in water, thus making impossible to acquire NMR spectra in solution. On the other hand, NMR analyses possibly acquired in solid phase would have provided very complex spectra providing not useful information. Useful information about the structural and chemical composition of crosslinked compounds, in terms of functional groups present in the composites, has been instead furnished by the ATR-FTIR spectra which have been supplied.

 The authors stated that “ Due to these characteristics that well fit within the requirements for cell adhesion, infiltration, and proliferation, aerogel-like composites M4, M5, M6 and M9 could be suitable candidates for applications in TE and regenerative medicine.” No in vivo cell experiments were performed to prove it. Please provide the necessary set of experiments – cell viability, proliferation, differentiation, etc.

We apologize in advance to the Reviewer, but experiments on cells are usually in vitro and not in vivo. Anyway, we have not included here the experiments required by the Reviewer because out of scope of the present paper. As reported in the title and in the conclusions (lines 863-865), the scope of the present work was synthetizing a series of aerogels by crosslinking Gel B applying our novel protocol and carrying out their complete physicochemical characterization to detect the most promising ones to be further studied in the biological experiments required by the Reviewer. In our opinion, there is no place for biological experiments in this study, which appears already very complex. We assure the Reviewer, that such experiments are currently ongoing, and results will be reported in a subsequent biologic work, as already reported in the conclusions of original form of the manuscript (lines 885-886, revised manuscript). For more clarity, further specifications on this question have been added in the abstract.

There is more than 15% self-citation (53, 54, 55, 56, 63, 67, 68, 77, 82, 87, 88, 89, 90, 91, 92). Please consider decreasing it.

Inserting additional and more updated references and other references to meet the requests or comments by other Reviewers, the percentage of self-citation has been reduced.

Reviewer 3 Report

Comments and Suggestions for Authors

The manuscript “Synthesis and Physicochemical Characterization of Gelatine-Based Biodegradable Aerogel-Like Composites as Possible Scaffolds for Regenerative Medicine” by Alfei et al. presents an extensive study on the physicochemical properties of a scaffold developed by enzymatic cross-linking of gelatine B. The authors suggest that the fabricated aerogel scaffold could be a potential candidate for tissue engineering. The effort shows that the authors have gone an extra mile to include all the possible characterization profiles for the aerogel that can prove its suitability in tissue engineering and regenerative medicine. That being said, biocompatibility of a biomaterial scaffold should be considered as the gold standard to demonstrate its appropriateness for tissue engineering.

Although the results for water absorption, degradation, rheology look promising, the manuscript is completely based on assumptions. The use of phrases such as “…can be considered, could be the choice, could be suitable, could give cells more time to proliferate, could allow its easier reabsorption…”, shows lack of confidence and uncertainty. It shows that there is a 50-50 chance that it may or may not work. Secondly, the abstract is entirely misleading which points out words like “biocompatibility, adhesion, infiltration, proliferation” which gives an impression that a considerable amount of cell culture studies have been presented in the manuscript, which is not the case. Thirdly, too many abbreviations makes it hard for the readers to follow the results and conclusions. A schematic diagram of the whole cross-linking process would have been easy on the eyes of the reader.

Other minor comments are:

1.     How did the authors ensure the removal of H2O2 released during the cross-linking procedure? Since they have not performed any cell culture experiments, how did they confirm that the presence of H2O2 will not have an adverse effect on the mammalian cells.

2.     Line 399, what is NCF/collagen?

3.     Line 404, apparent density is reported in table 4, please rectify.

4.     A digital image of the formed aerogel would have been helpful in providing a clear picture of the appearance of the aerogel.

5.     WAC% is mentioned for the first time in the text in line 304 and its full form is given in line 381.

6.     Line 459, what are COB scaffolds?

7.   Line 748, black should be corrected to blank.

Author Response

The manuscript “Synthesis and Physicochemical Characterization of Gelatine-Based Biodegradable Aerogel-Like Composites as Possible Scaffolds for Regenerative Medicine” by Alfei et al. presents an extensive study on the physicochemical properties of a scaffold developed by enzymatic cross-linking of gelatine B. The authors suggest that the fabricated aerogel scaffold could be a potential candidate for tissue engineering. The effort shows that the authors have gone an extra mile to include all the possible characterization profiles for the aerogel that can prove its suitability in tissue engineering and regenerative medicine. That being said, biocompatibility of a biomaterial scaffold should be considered as the gold standard to demonstrate its appropriateness for tissue engineering.

Although the results for water absorption, degradation, rheology look promising, the manuscript is completely based on assumptions. The use of phrases such as “…can be considered, could be the choice, could be suitable, could give cells more time to proliferate, could allow its easier reabsorption…”, shows lack of confidence and uncertainty. It shows that there is a 50-50 chance that it may or may not work. Secondly, the abstract is entirely misleading which points out words like “biocompatibility, adhesion, infiltration, proliferation” which gives an impression that a considerable amount of cell culture studies have been presented in the manuscript, which is not the case. Thirdly, too many abbreviations makes it hard for the readers to follow the results and conclusions. A schematic diagram of the whole cross-linking process would have been easy on the eyes of the reader.

We thank the Reviewer for his/her comments, that will be met when our project of research in TE will be complete. Being the whole work vast, the progressive results obtained cannot be reported in a single paper. We have decided to divide the project in three phases, and to report the related findings in three papers. In the first phase, a new protocol to crosslink gelatine without recovering to the commonly used crosslinking agents but mimicking the enzymatic natural process of collagen crosslinking, was developed. In this first phase, it was necessary designing and performing several syntheses and a hard work of purification and characterization of monomers and copolymers to be used to crosslink gelatine. Moreover, tests were necessary to select the copolymers really substrates of lysyl oxidase (LO) and then tests of Gel B reticulation to optimize the results. All this work has been reported in our recent paper (Ref. 8, revised manuscript). In the second phase, whose results have been reported in the present paper, we employed the developed protocol, further optimized, to prepare up to nine samples which needed an as complete as possible physicochemical characterization to select those possessing at least the physicochemical requisites to be successful in future biological experiments, including cells adhesion, infiltration, and proliferation. In fact, as reported in the title and in the conclusions (lines 863-865) the scope of the present work was synthetizing a series of aerogels by crosslinking Gel B and carrying out their complete physicochemical characterization to detect the most promising ones to be further studied in the biological experiments required by the Reviewer. There is no place for biological experiments in this study, which appears already very complex. This work could have already contained biological experiments if only one scaffold had been prepared, thus requiring fewer characterization experiments since the choice of the compound to send to the biological tests would have been forced. As reported in the conclusion section (lines 885-886), biological experiments are currently ongoing, and results will be reported in a subsequent biologic work. Anyway, for more clarity and less misleading issues, further specifications on this question have been added in the abstract. Concerning the Schematic procedure of the crosslinking process, it was already inserted in the original version of the manuscript as Scheme 1.

Other minor comments are:

  1. How did the authors ensure the removal of H2O2 released during the cross-linking procedure? Since they have not performed any cell culture experiments, how did they confirm that the presence of H2O2 will not have an adverse effect on the mammalian cells.

As reported in the experimental section, crosslinked materials have been purified by dialysis and subsequently lyophilized. The dried residues were further extracted with H2O or DMSO and then lyophilized again. We are confident that upon these purification procedures, residuals of H2O2, which is a molecule highly instable, could not be possible. Anyway, differently from what asserted by the Reviewer, we have not reported any confirmation about the possible positive or adverse effects of our scaffolds on cells, because such reports will be possible only upon the due experiments which are out of scope of the present paper. For what we know at the moment, our scaffold could trigger oxidative stress on cells in a manner not dependent on the residual H2O2 supposed by the Reviewer.

  1. Line 399, what is NCF/collagen?

NCF has been specified. Please, see lines 421-422.

  1. Line 404, apparent density is reported in table 4, please rectify.

The Reviewer request has been satisfied. Please, see line 427.

  1. A digital image of the formed aerogel would have been helpful in providing a clear picture of the appearance of the aerogel.

A digital image has been included in the Supplementary Materials as Figure S1.

  1. WAC% is mentioned for the first time in the text in line 304 and its full form is given in line 381.

We thank the Reviewer for his/her attention. In the revised version of the manuscript WAC% has been explained at its first mention. Please, see line 325.

  1. Line 459, what are COB scaffolds?

As asked, COB has been specified. Please, see line 481.

  1. Line 748, black should be corrected to blank.

We thank a lot the Reviewer. The Reviewer request has been satisfied in line 773.

Round 2

Reviewer 2 Report

Comments and Suggestions for Authors

Even the biological in vitro studies are very important, I accept the author's responses to my suggestions, and the manuscript can be published in the latest form.

Reviewer 3 Report

Comments and Suggestions for Authors

Authors have very well incorporated all the comments into their work. No such errors or inconsistencies were noticed.